# Computational Design and Experimental Evaluation of MERS-CoV siRNAs in Selected Cell Lines

**DOI:** 10.3390/diagnostics13010151

**Published:** 2023-01-02

**Authors:** Sayed S. Sohrab, Sherif A. El-Kafrawy, Zeenat Mirza, Ahmed M. Hassan, Fatima Alsaqaf, Esam I. Azhar

**Affiliations:** 1Special Infectious Agents Unit, King Fahd Medical Research Center, King Abdulaziz University, Jeddah 21589, Saudi Arabia; 2Department of Medical Laboratory Sciences, Faculty of Applied Medical Sciences, King Abdulaziz University, Jeddah 21589, Saudi Arabia; 3King Fahd Medical Research Center, King Abdulaziz University, Jeddah 21589, Saudi Arabia

**Keywords:** MERS-CoV, in silico prediction, siRNAs, Vero cells, HEK-293-T, Huh-7 cells

## Abstract

Middle East respiratory syndrome coronavirus (MERS-CoV) is caused by a well-known coronavirus first identified in a hospitalized patient in the Kingdom of Saudi Arabia. MERS-CoV is a serious pathogen affecting both human and camel health globally, with camels being known carriers of viruses that spread to humans. In this work, MERS-CoV genomic sequences were retrieved and analyzed by multiple sequence alignment to design and predict siRNAs with online software. The siRNAs were designed from the orf1ab region of the virus genome because of its high sequence conservation and vital role in virus replication. The designed siRNAs were used for experimental evaluation in selected cell lines: Vero cells, HEK-293-T, and Huh-7. Virus inhibition was assessed according to the cycle threshold value during a quantitative real-time polymerase chain reaction. Out of 462 potential siRNAs, we filtered out 21 based on specific selection criteria without off-target effect. The selected siRNAs did not show any cellular toxicity in the tested cell lines at various concentrations. Based on our results, it was obvious that the combined use of siRNAs exhibited a reduction in MERS-CoV replication in the Vero, HEK-293-T, and Huh-7 cell lines, with the highest efficacy displayed in the Vero cells.

## 1. Introduction

Coronaviruses are serious pathogens responsible for respiratory diseases in mammals. In 2012, a new coronavirus was identified in an infected butcher in the Kingdom of Saudi Arabia, who developed a respiratory illness and died 11 days after hospital admission [1]. Based on the genomic and pathogenic character of the virus, it was ultimately designated as Middle East respiratory syndrome coronavirus (MERS-CoV), which was the sixth known pathogenic human coronavirus after SARS to infect both animals and humans. Since then, MERS-CoV has been reported in 27 countries, with 2583 cases and 889 deaths (https://www.who.int/, last accessed on 2 June 2022), becoming a global issue for human and camel health [1,2,3]. Camels are considered MERS-CoV reservoirs and the primary source of human infection. Additionally, camel workers are an intermediary source of viral transmission to other communities. However, it has been observed that most MERS-CoV patients have no camel contact, making the source of the infection uncertain. The possible role of camels in disease transmission to humans has been investigated in various locations [1,2,3,4]. The most common symptoms include fever and shortness of breath, but more serious cases develop multiorgan failure [5,6]. 

The MERS-CoV belongs to the family *Coronaviridae.* The size of the viral genome is approximately 25–32 kb. MERS-CoV falls under lineage C betacoronaviruses (βCoVs). To date, only four coronavirus groups have been reported—designated as Alpha, Beta, Gamma, and Delta—and they have high genetic diversity, which favors the emergence of new strains [7]. Genetic analyses have concluded that MERS-CoV emerged after the exchange of genetic materials from bats or camels to humans [8]. Currently, no USFDA-approved vaccines or potential therapy has been reported for MERS-CoV, but various approaches are being used to develop vaccines and antivirals. Oligonucleotide-based antiviral therapies using siRNA, shRNA, and miRNA are being evaluated. Long noncoding RNAs (lncRNAs) against cancers [9,10], bacterial infections [11], fungal infections [12,13], parasitic infections [14,15], viral infections [16]. Several potential RNA interference-based (RNAi) drugs have recently been reported [17]. Additionally, as an adaptive immune system, CRISPR-Cas is currently used to treat many diseases caused by viruses and bacteria [18]. The details of CRISPR technology and its effective uses against viral infections have recently been reviewed and reported [19,20,21,22]. The computational prediction of siRNAs and their in vitro evaluation against respiratory viruses have also been reported [23,24,25,26,27,28,29]. Additionally, oligonucleotide-based (miRNA and siRNA) antiviral therapies are being evaluated in clinical trials [30] against many different viruses, including the Flock House virus (FHV) [31], dengue virus [32], hepatitis C virus (HCV) [33], influenza virus [34,35], hepatitis B virus (HBV) [36], human papillomavirus (HPV) [37], SARS coronavirus (SARS-CoV) [38], and MERS-CoV [28,29]. 

This present work gains significance because of the lack of antiviral therapy or vaccines for MERS-CoV, and the generated data will provide valuable information about the advancement and further use of oligonucleotide-based antivirals as an alternative therapy. The main objectives of this work were to design potential siRNAs by using a computational approach and to experimentally evaluate the reduction of viral load in Vero, HEK-293-T, and Huh-7 cell lines. Based on the results, a significant reduction in viral RNA was observed after using the combined siRNAs.

## 2. Materials and Methods

### 2.1. Sequence Analysis and siRNAs Selection

The MERS-CoV genomic sequences were retrieved from GenBank (KF958702, KT368879) and aligned by using BioEdit software (Version 7.2). The orf1ab region was identified as a potential target during siRNA design. The siRNAs were designed and selected by siDirect (Version 2.0) [25,26,28,29,39,40]. 

### 2.2. Secondary Structure Prediction of siRNAs 

An online bioinformatics tool (RNAfold server—http://rna.tbi.univie.ac.at accessed on 1 May 2022) was used to predict the secondary structure of the siRNAs. We have provided figures of the predicted siRNAs in our manuscript. 

### 2.3. Cell Culture, Virus Propagation, and Virus Titering

Dulbecco’s modified Eagle’s medium (DMEM) was used to grow the cells at 37 °C. The growth and quantification of MERS-CoV infection as well as viral RNA products of replication were performed by using standard published protocols [4,41]. The siRNAs were delivered by Lipofectamine 2000 (Invitrogen, Waltham, MA, USA) to the grown cells (1 × 10^4^) with 60–80% confluency. All the experiments were performed in triplicate. The antiviral potency of the siRNAs was accessed 48 h post-transfection. We selected these three cell lines (Vero, HEK-293-T, and Huh-7) based on their differential cell line susceptibility, replicative capacity, better cell growth and multiplication, and their cytopathic effects in in vitro assays [42,43,44]. 

### 2.4. siRNA Transfection and Cytotoxicity Assay 

The siRNAs were delivered to the grown cells by using reverse transfection with Lipofectamine 2000 (Invitrogen, Waltham, MA, USA) following the manufacturer’s instructions. The cells were procured from ATCC^®^ and further grown in a standard culture plate at defined conditions in DMEM. The standard siRNA dilutions were prepared at various concentrations (50, 25, 10, and 5 nM) from 50 μM stock solutions by adding 100 μL of Opti-MEM and Lipofectamine 2000 and further incubated at room temperature for 30 min. The complex mixture was delivered slowly to the grown cells, which were allowed to grow further for 24 h at 37 °C. The cellular toxicity of the siRNAs in different cell lines was analyzed by using an MTT assay kit (Invitrogen) following the manufacturer’s instructions. The absorbance was measured at 570 nm using a SpectraMax i3x imaging cytometer, and the standard formula was applied for the cytotoxicity calculation. The detailed protocol for the cytotoxicity assay has been described previously [29].

### 2.5. Evaluation of Virus Replication Inhibition by qRT-PCR

To evaluate the inhibitory effect of siRNAs against MERS-CoV experimentally, we used multiple combinations of siRNAs with various doses (50, 25, 10, and 5 nM) on selected grown cells. In our earlier study, we evaluated the siRNAs at various concentrations ranging from 0.1, 0.5, 1.0, 2.5, 5.0, 10, 25, and 50 nM. Here, the concentrations of siRNAs (50, 25, 10, and 5 nM) were chosen because they produced better results than other concentrations in our previous study [29,45]. The grown cells were siRNA-transfected for 48 h and then inoculated with MERS-CoV at a multiplicity of infection (MOI) of 0.01, following the published protocol from our lab [1]. The inoculated cell lines were allowed to grow in appropriate conditions, and the cytopathic effect in all cell lines was observed daily for 72 h. The cell supernatant and lysate were separated from all the tested samples. Grown cells without siRNA transfection and without virus infection were used as a negative control, and the cells with virus infection were used as a positive control in triplicate. The purification of viral RNA was performed by using a commercial QIAamp Viral RNA Mini Kit (Qiagen, Germantown, MD, USA). The inhibition of virus replication was measured by determining the cycle threshold (Ct) value during real-time polymerase chain reaction (qPCR). The idea behind using the supernatant and lysate from the cell lines was to access the differences in Ct value among the tested cells, as some cells are still attached to the surface of culture plates; therefore, it is expected that there will be variation among the Ct values of the cell lines. The RNA was subjected to qPCR using MERS-CoV-specific primers [1]. The Ct value was used to analyze the inhibitory effect of each siRNA in both cell supernatant as well as cell lysate at various concentrations (50, 25, 10, and 5 nM). 

## 3. Results

### 3.1. Sequence Analysis and Prediction of siRNAs 

The multiple sequence alignment results showed high conservation at various locations among the human and camel isolates. The conserved region of the ORF1ab gene is presented in Figure 1. In this experiment, we selected only 21 out of 462 siRNAs, which were generated by siDirect software (Version 2.0) without any off-targets or similarities with any human mRNA sequences following the basic criteria. Table 1 provides the sequence, minimum free energy, and position of the designed siRNAs.

### 3.2. Cytotoxicity Assay

The cellular toxicities of the designed siRNAs were evaluated in the three selected cell lines and were observed to be concentration-dependent. None of the tested siRNAs displayed significant toxicity in any cell lines. The data were statistically analyzed by using the GraphPad Prism (9.3.0) two-way ANOVA software and considered significant with a *p*-value of <0.0001. The cytotoxicity results of each cell line are provided in Table 2 and Figure 2. 

### 3.3. Analysis of Virus Inhibition

The virus inhibition and the level of viral RNA in the cell lines were determined by the Ct value of qPCR for all combinations of siRNAs in the cell supernatant as well as the cell lysate. The analysis was performed by using only four different concentrations (50, 25, 10, and 5 nM) of siRNAs, as per previous works [28,29]. The viral inhibition was observed to be dose-dependent for different siRNA combinations in Vero cells, HEK-293-T cells, and Huh-7 cells at various concentrations. In the Vero cell supernatant, the highest qPCR Ct value was observed for the siRNA 1+3 combination at all the concentrations, whereas the siRNA 1+8 combination showed a higher Ct value than the positive control at concentrations of 50 nM and 10 nM. The lowest Ct value was 13.17 among all tested combinations, and the highest was 20.90. The siRNA combinations 1+8 and 1+9 in the cell lysate showed a higher Ct value at a concentration of 5 nM as compared to the positive control (Table 3, Figure 3). In the HEK-293-T cell supernatant, the highest Ct value (35.17) was observed for the siRNA 1+6 combination at a concentration of 25 nM, whereas in the cell lysate, many siRNA combinations (1+2-25 nM, 1+3-5, 50 nM, 1+6-25 nM, 1+7-5 nM, and 1+8-5, 50 nM) showed higher Ct values at various concentrations (50–5 nM) as compared to the positive control (Table 4, Figure 4). Based on the in vitro results of all cell lines, it appeared that the combined use of different siRNAs was better for virus inhibition compared to using a single siRNA. In Huh-7 cells, the highest Ct value was observed in the supernatant of siRNA combinations 1+13 and 1+18 at concentrations of 50, 5, and 10 nM, and in the cell lysate, siRNA combinations 1+11 and 1+15 showed the highest Ct value at 10 nM and 5 nM. Interestingly, many siRNA combinations showed higher Ct values than the positive control (Table 5, Figure 5). The results for different siRNAs at the same concentrations is comparable because of the different cell lines and their replication as well as the growth and multiplication of the virus in the tested cell lines. 

### 3.4. Prediction of Secondary Structure of siRNA 

The RNAfold server was used to predict the secondary structure of the designed siRNAs. The minimum free energy (MFE) and thermodynamics ensemble (%) data for all siRNAs are provided in Table 1, and their secondary structures are presented in Figure 6. The secondary structure of the selected siRNA follows the selection criteria of the RNAfold and has no structure complexity. The prediction showed better binding efficiency to the target sequences with potential silencing of the target genes. 

## 4. Discussion 

In 2012, MERS-CoV was identified in a hospitalized patient in the Kingdom of Saudi Arabia, where the highest fatality rate of the disease has been reported. Tremendous efforts have been made in MERS-CoV research using varied approaches that have contributed significantly towards disease management. Despite this enormous global research effort, no vaccines or antivirals are currently available against MERS-CoV, but various strategies are being used for their development, including oligonucleotide (siRNAs/miRNAs)-based therapy [46,47,48,49,50,51]. Ongoing research has successfully solved most complications related to siRNAs-based therapy, resulting in progressive outcomes against many viral diseases [52,53,54]. Currently, only one siRNA known as ALNRSV01 has been approved by the WHO for human use [16]. Various siRNAs for MERS-CoV have been designed by computational approaches, but their antiviral efficacy has not been evaluated in vitro [23,24]. Many new siRNAs have been designed for HCV and MERS-CoV inhibition, and experimentally observed to be effective in the reduction of viral RNA load in selected cells [28,29,40]. Recently, Fukushige et al. used hyaluronic acid-coated liposomes as alternative liposomes for siRNA delivery to lung cells [55]. In our study, we have used the bioinformatics approach to design and evaluate the antiviral potency of MERS-CoV siRNAs. 

In our study, the software produced 462 siRNAs from the orf1ab region. We selected and evaluated only 21 siRNAs following the standard criteria for filtration [25,26]. The prediction of the siRNA secondary structure was performed by using the online RNAfold server, which was also used to calculate the MFE and partition function of the RNA by reading the RNA sequences. 

We evaluated only certain siRNAs for combined use in the Vero, Huh-7, and HEK-293-T cell lines. The cellular toxicity of each designed siRNA was evaluated in the three cell lines before inducing virus infection and qPCR. The results showed no cytotoxicity of the evaluated siRNAs in the tested cell lines. The experimental evaluation results of each siRNA combination and their antiviral potency were significantly variable across different cell lines, even at the same concentration, which could be due to the growth and multiplication properties of the cell lines after siRNA delivery and virus infection. Many siRNA combinations showed significant and strong inhibitory effects toward viral replication in Vero cells. The Ct values from the qPCR data indicated a significant inhibition of replication and a reduction in viral RNA in both the cell supernatant and lysate of the Vero, Huh-7, and HEK-293-T cell lines. We evaluated all 21 combinations of siRNAs in the selected cell lines. Better siRNA efficacy and reduction of viral load were observed in Vero cells due to the better growth and multiplication of the virus in this cell line. In silico approaches are commonly used to screen possible strategies for overcoming infections, with successful examples in similar studies that emphasize bioinformatic approaches and their utility [56]. One limitation of this study was that siRNA treatment was conducted in only selected cell lines because MERS-CoV does not grow and multiply in most other cell lines. Further evaluation of additional siRNA combinations in multiple cell lines is needed to use as oligonucleotide-based antivirals for MERS-CoV. The findings of this study should be further evaluated in mice and other human primates, which are lacking here in our facility.

## 5. Conclusions

Based on our computational approach combined with the experimental evaluation of selected siRNAs in terms of cytotoxicity and qPCR, we concluded that the in silico designing and filtration of siRNAs can be an effective approach to alternative oligonucleotide-based therapeutics for MERS-CoV. The experimental evaluation results of the antiviral potency of siRNAs provided valuable information about the selection of siRNAs tested in various cell lines at selected concentrations with multiple combinations. Recently, after many barriers and challenges were overcome, the FDA approved several siRNA-based therapeutics. However, many other challenges must still be addressed, such as enzymatic degradation, rapid renal clearance, endosomal trapping, plasma protein sequestration, and activation of the immune system. To overcome these barriers, many strategies and techniques are being used to modify the backbone, bases, sugars, conjugation of aptamers, and exosome-based delivery of siRNAs. We observed that the Vero cells were better than other cells because of virus multiplication and cell growth for testing the siRNAs against MERS-CoV. Finally, the overall results proved that the computational approach of this study can be used to design siRNAs as potential therapeutics against MERS-CoV, and our experimental evaluation provided information about the selection of these siRNAs. 

## Figures and Tables

**Figure 1 diagnostics-13-00151-f001:**
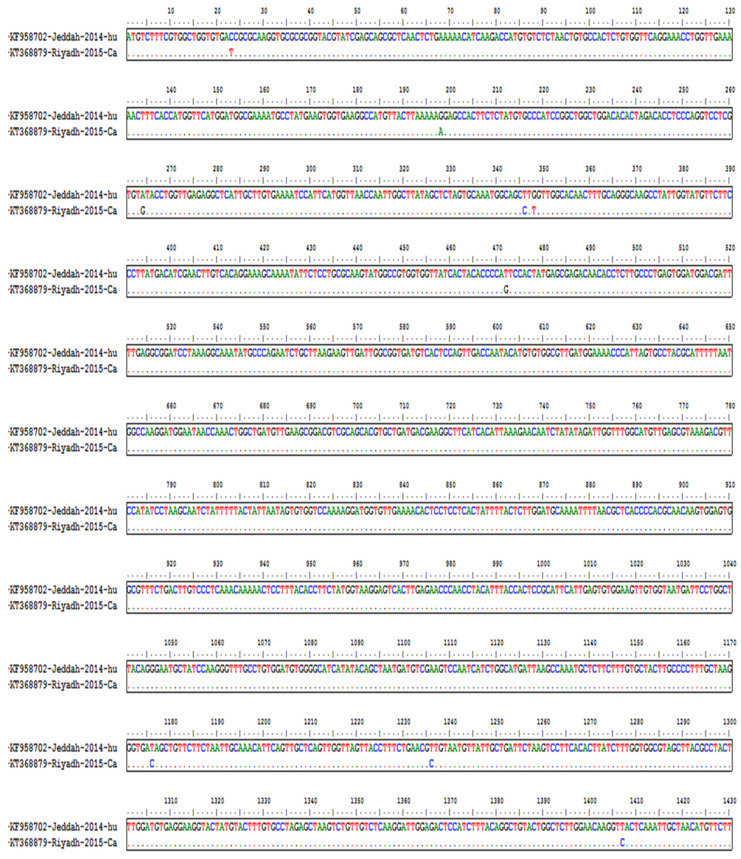
Multiple sequence alignment and homology of the MERS-CoV genome isolated from humans and camels. The figure shows good homology with very few variations in the viral genome.

**Figure 2 diagnostics-13-00151-f002:**
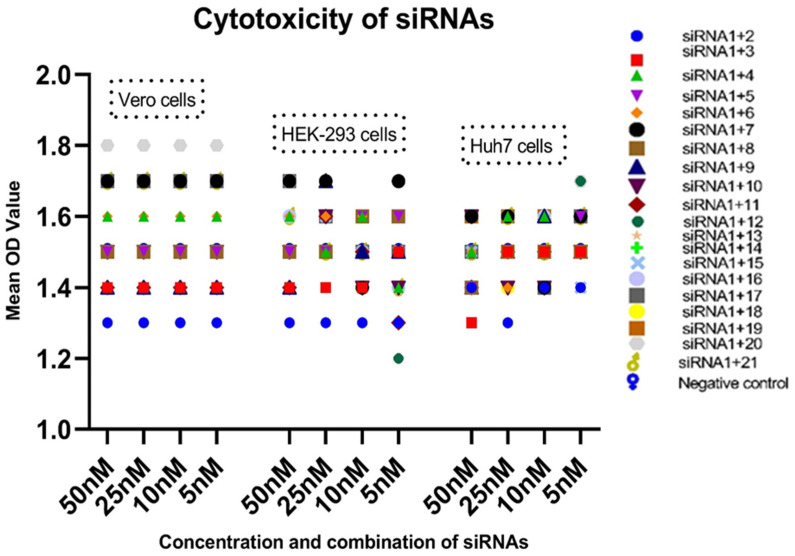
Cytotoxicity of siRNAs in selected cell lines at different concentrations. The variable concentrations of siRNA combinations (50, 25, 10, and 5 nM) were delivered to the grown cell lines. No cell lines showed significant cytotoxicity.

**Figure 3 diagnostics-13-00151-f003:**
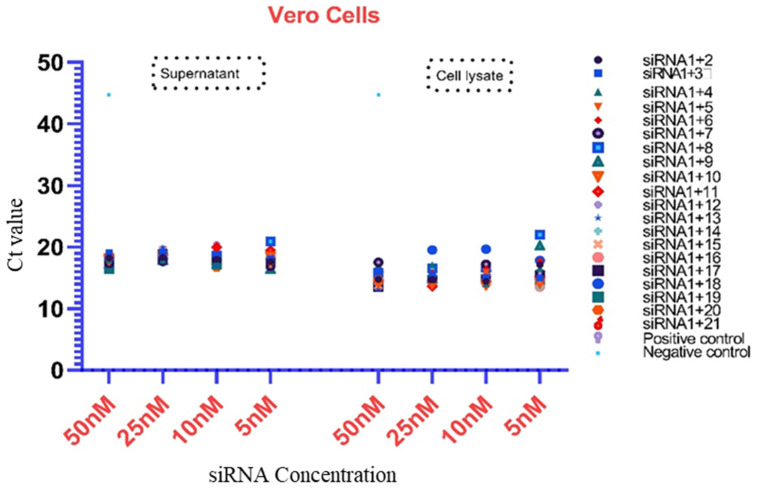
Presentation of the Ct values of the combined use of siRNAs in both the cell lysate and supernatant of Vero cells. Some siRNAs showed better potential as compared to other combinations (50, 25, 10, and 5 nM). The highest Ct value was 45, which occurred in the negative control sample.

**Figure 4 diagnostics-13-00151-f004:**
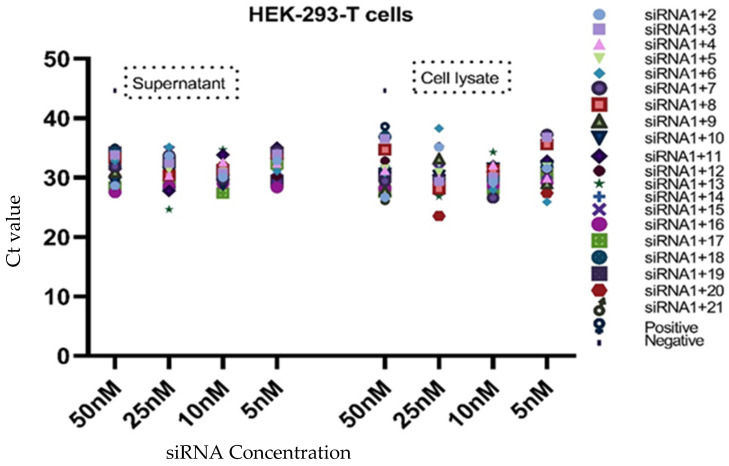
Presentation of the Ct values of combined use of siRNAs in both the cell lysate and supernatant of HEK-293-T cells. The graph shows that a few siRNAs exhibited better potential compared to other combinations (50,25, 10, and 5 nM). The highest Ct value was 45, which occurred in the negative control sample.

**Figure 5 diagnostics-13-00151-f005:**
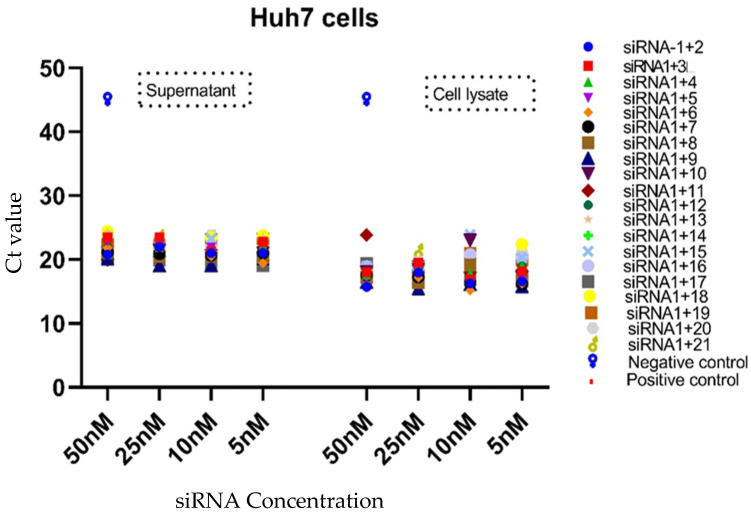
Presentation of Ct values of combined use of siRNAs in both the cell lysate and supernatant of Huh-7 cells. The multiple combinations of siRNAs were delivered to Huh-7 cells at different concentrations (50,25, 10, and 5 nM). The qPCR data were used to draw the graph using the graph Pad software. The data show the efficiency of the siRNAs at various combinations and concentrations. The highest Ct value was 45, which occurred in the negative control sample.

**Figure 6 diagnostics-13-00151-f006:**
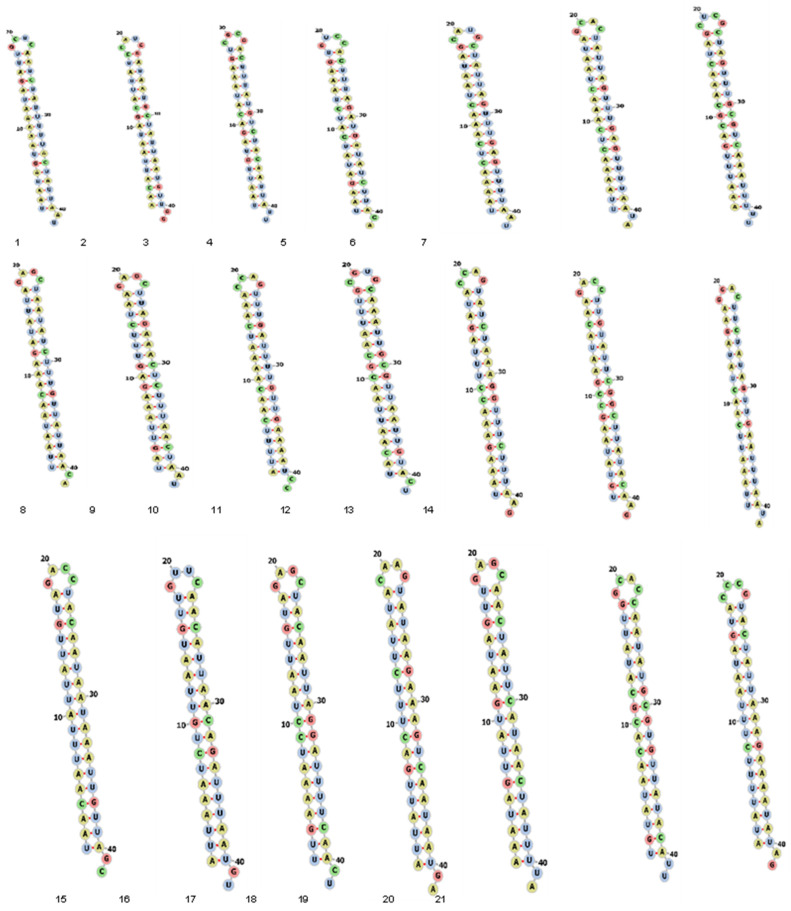
Predicted secondary structure of selected siRNAs from the orf1ab region.

**Table 1 diagnostics-13-00151-t001:** List of designed and filtered siRNAs from the orf1ab region (KF958702).

S.No.	Position of siRNA in the Genome (Start–End)	Target Sequence	Predicted RNA Oligo Sequences (5′→3′)	Minimum Free Energy (MFE (kcal/mol) and Frequency of Thermodynamic Ensemble (%)
1	791–813	agcaatctatttttactattaat	UAAUAGUAAAAAUAGAUUGCU CAAUCUAUUUUUACUAUUAAU	−17.96, 65.21
2	1615–1637	atggataatgctattaatgttgg	AACAUUAAUAGCAUUAUCCAU GGAUAAUGCUAUUAAUGUUGG	−21.80, 77.37
3	1910–1932	gcgactttatgtctacaattatt	UAAUUGUAGACAUAAAGUCGC GACUUUAUGUCUACAAUUAUU	−22.02, 69.74
4	4018–4040	gacactttagatgatatcttaca	UAAGAUAUCAUCUAAAGUGUC CACUUUAGAUGAUAUCUUACA	−22.62, 69.54
5	5597–5619	atgctattagtttgagttttaat	UAAAACUCAAACUAAUAGCAU GCUAUUAGUUUGAGUUUUAAU	−21.51, 83.64
6	5598–5620	tgctattagtttgagttttaata	UUAAAACUCAAACUAAUAGCA CUAUUAGUUUGAGUUUUAAUA	−19.74, 57.91
7	5819–5841	gagctagtttgcgtcaaattttt	AAAUUUGACGCAAACUAGCUC GCUAGUUUGCGUCAAAUUUUU	−24.28, 53.63
8	9495–9517	ctctaatatctttgttattaaca	UUAAUAACAAAGAUAUUAGAG CUAAUAUCUUUGUUAUUAACA	−17.97, 54.45
9	9533–9555	ctcttagaaactctttaactaat	UAGUUAAAGAGUUUCUAAGAG CUUAGAAACUCUUUAACUAAU	−22.37, 64.54
10	13,605–13,627	tggtttgattttgttgaaaatcc	AUUUUCAACAAAAUCAAACCA GUUUGAUUUUGUUGAAAAUCC	−18.34, 35.22
11	14,005–14,027	acgcaaattgcgttaattgtact	UACAAUUAACGCAAUUUGCGU GCAAAUUGCGUUAAUUGUACU	−22.34, 79.46
12	14,389–14,411	tggtatctaaaggtttctttaag	UAAAGAAACCUUUAGAUACCA GUAUCUAAAGGUUUCUUUAAG	−22.04, 67.95
13	16,177–16,199	gtcttgtattcggcttatacaag	UGUAUAAGCCGAAUACAAGAC CUUGUAUUCGGCUUAUACAAG	−26.53, 58.68
14	16,217–16,239	tccttctatagttgaatttaata	UUAAAUUCAACUAUAGAAGGA CUUCUAUAGUUGAAUUUAAUA	−20.24, 48.81
15	17,283–17,305	gtctacaataataaattgttagc	UAACAAUUUAUUAUUGUAGAC CUACAAUAAUAAAUUGUUAGC	−17.87, 75.42
16	17,583–17,605	aacaacattaacagatttaatgt	AUUAAAUCUGUUAAUGUUGUU CAACAUUAACAGAUUUAAUGU	−19.59, 62.23
17	18,028–18,050	ctctacaattaggattttcaact	UUGAAAAUCCUAAUUGUAGAG CUACAAUUAGGAUUUUCAACU	−22.08, 53.94
18	19,806–19,828	ttgtataagaaagtcaataatga	AUUAUUGACUUUCUUAUACAA GUAUAAGAAAGUCAAUAAUGA	−19.97, 64.53
19	20,090–20,112	ctcaactattcataactatttta	AAAUAGUUAUGAAUAGUUGAG CAACUAUUCAUAACUAUUUUA	−19.63, 42.01
20	20,498–20,520	tgccaatatgcgtgttatacatt	UGUAUAACACGCAUAUUGGCA CCAAUAUGCGUGUUAUACAUU	−25.98, 74.21
21	20,948–20,970	gggtactattaaagaaaatatag	AUAUUUUCUUUAAUAGUACCC GUACUAUUAAAGAAAAUAUAG	−17.65, 66.76

**Table 2 diagnostics-13-00151-t002:** Cytotoxicity (CC50) of siRNAs in selected cells.

siRNAs Combination	siRNA Concentrations (nM)/OD Value in Vero Cells					siRNA Concentrations (nM)/OD Value in HEK-293-T Cells						siRNA Concentrations (nM)/OD Value in Huh7 Cells			
	50	25	10	5.0	CC50	50	25	10	5.0	CC50	50	25	10	5.0	CC50
siRNA-1+2	1.3	1.3	1.3	1.3	>100	1.3	1.3	1.3	1.3	>100	1.4	1.3	1.4	1.4	>100
siRNA-1+3	1.4	1.4	1.4	1.4	>100	1.4	1.4	1.4	1.5	>100	1.3	1.5	1.5	1.5	>100
siRNA-1+4	1.6	1.6	1.6	1.6	>100	1.6	1.5	1.6	1.4	>100	1.5	1.6	1.6	1.5	>100
siRNA-1+5	1.5	1.5	1.5	1.5	>100	1.5	1.5	1.6	1.6	>100	1.4	1.5	1.5	1.6	>100
siRNA-1+6	1.6	1.6	1.6	1.6	>100	1.6	1.6	1.6	1.5	>100	1.5	1.4	1.5	1.5	>100
siRNA-1+7	1.7	1.7	1.7	1.7	>100	1.7	1.7	1.4	1.7	>100	1.6	1.6	1.4	1.6	>100
siRNA-1+8	1.5	1.5	1.5	1.5	>100	1.5	1.5	1.6	1.6	>100	1.4	1.5	1.5	1.5	>100
siRNA-1+9	1.4	1.4	1.4	1.4	>100	1.4	1.7	1.5	1.5	>100	1.4	1.5	1.6	1.5	>100
siRNA-1+10	1.5	1.5	1.5	1.5	>100	1.5	1.6	1.4	1.4	>100	1.6	1.4	1.4	1.5	>100
siRNA-1+11	1.4	1.4	1.4	1.4	>100	1.4	1.6	1.5	1.3	>100	1.4	1.5	1.5	1.6	>100
siRNA-1+12	1.5	1.5	1.5	1.5	>100	1.5	1.4	1.3	1.2	>100	1.6	1.3	1.4	1.7	>100
siRNA-1+13	1.4	1.4	1.4	1.4	>100	1.6	1.4	1.5	1.5	>100	1.5	1.4	1.5	1.6	>100
siRNA-1+14	1.5	1.5	1.5	1.5	>100	1.5	1.6	1.5	1.4	>100	1.4	1.5	1.6	1.5	>100
siRNA-1+15	1.5	1.5	1.5	1.5	>100	1.7	1.6	1.5	1.5	>100	1.5	1.6	1.5	1.4	>100
siRNA-1+16	1.5	1.5	1.5	1.5	>100	1.6	1.5	1.5	1.4	>100	1.6	1.6	1.6	1.5	>100
siRNA-1+17	1.7	1.7	1.7	1.7	>100	1.7	1.6	1.5	1.5	>100	1.5	1.5	1.4	1.6	>100
siRNA-1+18	1.5	1.5	1.5	1.5	>100	1.5	1.6	1.5	1.4	>100	1.4	1.4	1.5	1.5	>100
siRNA-1+19	1.7	1.7	1.7	1.7	>100	1.7	1.6	1.5	1.5	>100	1.6	1.6	1.6	1.6	>100
siRNA-1+20	1.8	1.8	1.8	1.8	>100	1.5	1.5	1.4	1.3	>100	1.4	1.4	1.5	1.7	>100
siRNA-1+21	1.7	1.7	1.7	1.7	>100	1.6	1.5	1.5	1.4	>100	1.5	1.6	1.5	1.6	>100
Negative	1.5	1.5	1.5	1.5	>100	1.5	1.5	1.5	1.5	>100	1.5	1.5	1.5	1.5	>100

**Table 3 diagnostics-13-00151-t003:** Ct value of qPCR of siRNA in Vero cells.

siRNAs Combinations	Vero Cells							
	(Cell Supernatant)—(nM)				(Cell Lysate)—(nM)			
	50	25	10	5.0	50	25	10	5.0
siRNA1+2	18.25	18.31	18.01	17.64	14.75	14.74	14.43	17.13
siRNA1+3	19.07	19.24	18.68	18.08	16.06	15.48	15.06	14.99
siRNA1+4	17.87	17.89	17.66	17.77	15.32	17.11	14.06	16.59
siRNA1+5	17.41	17.76	17.97	17.13	14.80	14.44	15.87	16.53
siRNA1+6	17.83	18.99	19.68	17.82	15.61	15.89	16.43	17.74
siRNA1+7	17.34	17.80	17.97	16.87	17.50	14.86	17.22	15.75
siRNA1+8	17.48	17.85	18.58	20.90	15.82	16.56	16.63	21.96
siRNA1+9	17.25	17.84	17.91	16.46	16.62	14.88	16.76	20.23
siRNA1+10	18.24	18.14	18.06	18.46	14.09	13.96	13.78	14.11
siRNA1+11	18.59	18.10	19.97	19.46	14.46	13.60	14.33	15.21
siRNA1+12	18.08	17.71	20.45	18.24	14.62	14.28	14.80	14.61
siRNA1+13	18.26	19.97	18.07	18.63	13.17	14.50	14.15	15.60
siRNA1+14	18.32	18.63	18.02	17.42	15.19	14.31	14.38	13.48
siRNA1+15	18.32	19.61	19.68	19.12	13.67	14.41	15.07	14.76
siRNA1+16	17.90	18.06	18.97	18.71	14.73	14.01	14.11	13.50
siRNA1+17	17.64	18.91	18.09	17.97	13.52	14.41	14.88	15.48
siRNA1+18	17.20	17.67	17.66	17.12	15.75	19.54	19.67	17.81
siRNA1+19	16.42	17.81	17.10	17.48	14.51	14.78	14.76	14.59
siRNA1+20	18.48	17.68	16.67	17.97	15.27	15.17	15.14	15.35
siRNA1+21	17.86	18.97	18.25	18.37	14.88	14.94	15.31	14.70
Positive control	17.85				16.04			
Negative control	45.00				45.00			

**Table 4 diagnostics-13-00151-t004:** Ct value of qPCR results of siRNA in HEK-293-T cells.

siRNAs Combinations	HEK-293-T Cells							
	(Cell Supernatant)—(nM)				(Cell Lysate)—(nM)			
	50	25	10	5.0	50	25	10	5.0
siRNA1+2	28.62	33.66	30.06	32.81	26.69	35.10	29.21	31.55
siRNA1+3	33.84	32.42	30.77	33.96	36.58	29.32	30.10	36.76
siRNA1+4	33.88	30.34	32.63	32.44	31.20	29.89	32.14	29.89
siRNA1+5	33.56	31.24	30.92	32.94	31.40	30.83	29.02	31.75
siRNA1+6	32.86	35.17	29.87	31.10	30.78	38.28	27.96	25.90
siRNA1+7	31.87	33.71	29.73	34.48	29.42	29.78	26.60	37.23
siRNA1+8	33.23	30.37	31.35	34.04	34.71	28.11	31.24	35.56
siRNA1+9	30.88	30.37	31.35	34.04	27.77	33.15	31.82	29.01
siRNA1+10	29.35	30.92	28.85	34.17	30.72	30.32	31.60	31.73
siRNA1+11	31.44	27.76	33.83	35.02	30.86	30.35	31.96	32.82
siRNA1+12	28.58	31.44	31.36	30.21	32.84	35.20	30.61	31.23
siRNA1+13	30.60	24.63	34.69	32.46	29.46	26.86	34.27	28.80
siRNA1+14	32.03	31.94	31.72	33.09	31.97	31.17	30.59	31.96
siRNA1+15	29.57	34.56	30.26	29.91	29.91	31.45	31.53	30.58
siRNA1+16	27.51	28.53	29.00	28.33	28.15	29.48	28.59	29.14
siRNA1+17	28.42	29.69	27.40	32.28	27.54	29.25	28.12	30.53
siRNA1+18	34.18	28.25	29.47	31.94	36.87	27.93	29.93	31.84
siRNA1+19	34.23	32.65	30.54	29.49	30.31	29.67	29.66	31.06
siRNA1+20	30.28	28.46	31.86	29.04	28.49	23.53	31.28	27.38
siRNA1+21	29.06	29.94	30.36	29.46	26.61	28.42	28.77	29.66
Positive control	34.50				38.11			
Negative control	45.00				45.00			

**Table 5 diagnostics-13-00151-t005:** Ct values of qPCR results of siRNA combinations in Huh-7 cells.

siRNAs Combination	Huh-7 Cells
(Cell Supernatant)—(nM)	(Cell Lysate)—(nM)
50	25	10	5.0	50	25	10	5.0
siRNA1+2	20.77	21.97	21.02	21.07	15.71	17.99	16.25	16.60
siRNA1+3	23.42	23.54	21.31	22.77	18.04	19.54	16.90	18.17
siRNA1+4	20.77	21.94	21.22	22.20	17.68	19.07	17.89	19.03
siRNA1+5	22.82	22.10	21.86	21.87	17.05	19.00	16.72	17.08
siRNA1+6	21.80	22.59	20.53	19.43	17.55	17.17	15.19	16.22
siRNA1+7	21.15	20.91	20.66	21.00	17.57	17.27	16.32	16.18
siRNA1+8	21.24	20.44	20.53	20.41	17.21	16.41	19.15	16.93
siRNA1+9	20.10	18.97	18.96	20.09	16.42	15.37	16.06	15.73
siRNA1+10	21.17	21.51	20.68	21.12	18.11	18.50	23.11	17.25
siRNA1+11	19.99	20.48	21.10	19.86	23.87	17.88	18.02	18.29
siRNA1+12	22.03	22.49	19.85	22.70	17.92	18.72	18.48	18.97
siRNA1+13	24.33	20.02	21.31	23.48	18.85	18.60	18.03	19.05
siRNA1+14	23.53	22.03	20.95	22.26	18.42	19.12	18.66	17.26
siRNA1+15	20.81	22.31	23.25	22.65	18.97	19.02	23.94	20.26
siRNA1+16	21.06	22.05	21.95	20.61	18.90	18.74	20.82	20.40
siRNA1+17	19.99	19.93	19.36	19.05	19.38	18.93	17.12	16.61
siRNA1+18	24.39	21.99	23.44	23.74	18.36	18.17	18.60	22.40
siRNA1+19	22.36	22.17	23.18	23.26	18.10	17.89	20.95	19.88
siRNA1+20	23.97	23.39	23.87	22.92	18.62	19.69	20.76	21.18
siRNA1+21	22.89	23.42	23.36	22.98	18.86	21.25	19.98	19.73
Positive control	21.10				19.18			
Negative control	45.00				45.00

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
