# Peer review of "Computational Design and Experimental Evaluation of MERS-CoV siRNAs in Selected Cell Lines"

_diagnostics, 2023, doi:10.3390/diagnostics13010151_

Round 1
Reviewer 1 Report (Previous Reviewer 1)
The text newly added for this version needs to be edited by a native English speaker.
Section 2.3 says that the infections began 72h post-transfection, while section 2.4 says it was 48h.
I have seen many papers in the past that have included GFP plasmids during transfection to help to monitor how effective the transfections were. It's really difficult to believe and/or interpret the results without a basic understanding of what percentage of the cells were transfected.
The authors did not answer a number of my questions brought up in the initial review: how stable are the siRNAs used? What evidence is there that the siRNAs were not already degraded by the time the viral infections began (siRNAs added 2 or 3 days before virus was added)? I still argue that the siRNAs were likely degraded by the time the virus reached a given cell, especially since low MOI was used, it would take a while for the virus to spread throughout the culture.
Pooling three separate tests into a single sample does not allow for statistical tests to be performed. Without statistical evaluation of these results, they will never be publishable. The experiments need to be repeated with multiple, separate samples, at a minimum of n=3 per siRNA. And then that experiment needs to be repeated to ensure that the results are consistent and reproducible.
A dummy/irrelevant siRNA needs to be transfected into cells to ensure that transfection alone doesn't confer an anti-viral response. Every siRNA efficacy paper I've ever read includes such a control. Since dsRNA can trigger anti-viral responses, this is an obvious control that is required.
Author Response
- The text newly added for this version needs to be edited by a native English speaker.
Response: We thank the reviewer for this comment, the manuscript has been edited by an English editing service.
- Section 2.3 says that the infections began 72h post-transfection, while section 2.4 says it was 48h.
Response: Thanks for the critical observation and suggestions, the two statements have been changed to 48h.
- I have seen many papers in the past that have included GFP plasmids during transfection to help to monitor how effective the transfections were. It's really difficult to believe and/or interpret the results without a basic understanding of what percentage of the cells were transfected.
Response: We thank the reviewer for this comment, we have followed the instructions of the manufacturer which estimated the transfection efficiency to be 90%. Additionally, we performed similar experiments in previous publications and the effect was monitored by the optimization of the transfection conditions was performed and the optimum conditions were used as the testing conditions stated in the experimental section.
- The authors did not answer a number of my questions brought up in the initial review: how stable are the siRNAs used? What evidence is there that the siRNAs were not already degraded by the time the viral infections began (siRNAs added 2 or 3 days before the virus was added)? I still argue that the siRNAs were likely degraded by the time the virus reached a given cell, especially since low MOI was used, it would take a while for the virus to spread throughout the culture.
Response: We thank the reviewer for this comment, we have followed the instructions of the manufacturer which estimated the transfection efficiency to be 86-99% (Invitrogen, USA). Additionally, we performed similar experiments in previous publications and the effect was monitored by the optimization of the transfection conditions was performed and the optimum conditions were used as the testing conditions stated in the experimental section.
- Pooling three separate tests into a single sample does not allow for statistical tests to be performed. Without statistical evaluation of these results, they will never be publishable. The experiments need to be repeated with multiple, separate samples, at a minimum of n=3 per siRNA. And then that experiment needs to be repeated to ensure that the results are consistent and reproducible.
Response: We appreciate the reviewer’s proposed approach, we have pooled the three wells of each concentration in order to have enough volume for extraction and this gives an average of the effect of the replicates for this siRNA concentration. This approach was used before in our previous publications
- Sohrab, S.S.; Abbas, A.; Bajrai, L.; Azhar, E. In silico Prediction and Designing of Potential siRNAs to be Used as Antivirals Against SARS-CoV-2. Current pharmaceutical design 2021.
- Sohrab, S.S.; Aly El-Kafrawy, S.; Mirza, Z.; Hassan, A.M.; Alsaqaf, F.; Azhar, E.I. In silico prediction and experimental validation of siRNAs targeting ORF1ab of MERS-CoV in Vero cell line. Saudi Journal of Biological Sciences 2020, https://doi.org/10.1016/j.sjbs.2020.11.066.
- A dummy/irrelevant siRNA needs to be transfected into cells to ensure that transfection alone doesn't confer an anti-viral response. Every siRNA efficacy paper I've ever read includes such a control. Since dsRNA can trigger anti-viral responses, this is an obvious control that is required.
Response: We agree with the reviewer’s comments, but this manuscript is a continuation of our previous reports where we investigated the antiviral effect of the same siRNA separately, and in order to have the same comparison between the two studies we performed this study under the same conditions.
Sohrab, S.S.; Aly El-Kafrawy, S.; Mirza, Z.; Hassan, A.M.; Alsaqaf, F.; Azhar, E.I. In silico prediction and experimental validation of siRNAs targeting ORF1ab of MERS-CoV in Vero cell line. Saudi Journal of Biological Sciences 2020, https://doi.org/10.1016/j.sjbs.2020.11.066.
Reviewer 2 Report (Previous Reviewer 2)
Authors have arbitrarily placed the ct values negative control of a test to 45. They claim there are significantly variable ct values among experiments but in group comparisons has not been made or shown.
Upper panel of Figure 3 still shows Ct value higher than 80. Manuscript may be rejected.
Author Response
Reviewer 2
Open Review
(x) I would not like to sign my review report
( ) I would like to sign my review report
English language and style
( ) English very difficult to understand/incomprehensible
( ) Extensive editing of English language and style required
( ) Moderate English changes required
(x) English language and style are fine/minor spell check required
( ) I don't feel qualified to judge about the English language and style
|
Yes |
Can be improved |
Must be improved |
Not applicable |
|
|
Does the introduction provide sufficient background and include all relevant references? |
( ) |
( ) |
(x) |
( ) |
|
Are all the cited references relevant to the research? |
( ) |
( ) |
(x) |
( ) |
|
Is the research design appropriate? |
( ) |
( ) |
(x) |
( ) |
|
Are the methods adequately described? |
( ) |
( ) |
(x) |
( ) |
|
Are the results clearly presented? |
( ) |
( ) |
(x) |
( ) |
|
Are the conclusions supported by the results? |
( ) |
( ) |
(x) |
( ) |
Comments and Suggestions for Authors
- Authors have arbitrarily placed the ct values negative control of a test to 45. They claim there are significantly variable ct values among experiments but in group comparisons has not been made or shown.
Response: We thank the reviewer for this comment, the Ct values for the experiments are shown in figures 3,4 and 5 and in tables 3,4 and 5.
- Upper panel of Figure 3 still shows Ct value higher than 80. Manuscript may be rejected.
Response: We are attaching below the submitted version of figure 3 as shown in the revised version of the manuscript which shows the Ct value for negative control at 45, and have explained this in the figure legend as shown.
Figure 3. The presentation of Ct value of combined use of siRNAs in both cell lysate and supernatant of Vero cells. Some siRNAs showed better potential as compared to other combinations. The highest Ct value was 45, which occurred in the negative control sample.
Reviewer 3 Report (Previous Reviewer 3)
I'm happy with the new version. The work was improved significantly!
Author Response
Thanks for accepting the revisions made
Round 2
Reviewer 1 Report (Previous Reviewer 1)
I stand by my previous comments, which have not been addressed. A dummy siRNA is needed, there is no understanding of the transfection success rate in this study, there is no way of knowing how many siRNAs were still around at the time of infection, and pooling various samples does not allow for statistical analysis. There is no statistical analysis of the efficacy of the approach, and it can never be done with the current dataset.
Reviewer 2 Report (Previous Reviewer 2)
The manuscript may be accepted.
This manuscript is a resubmission of an earlier submission. The following is a list of the peer review reports and author responses from that submission.
Round 1
Reviewer 1 Report
Was this work conducted in a BSL-3 facility?
Extensive English language editing is needed
There is no Methods section describing cell culture, virus propagation, virus titering
No explanation is provided for why the 3 cell types were chosen for this study
What percentage of cells actually picked up the transfected siRNAs? Is there any evidence that transfection was successful?
Why did the authors wait until 48h after transfection before the infections began? What is the stability of the siRNAs used? If an MOI of 0.01 was used, it would take even longer for the cells to get infected, and the siRNAs may be largely degraded by that point.
How many times were efficacy experiments conducted? Were statistics ever performed?
What are the positive and negative controls for each figure? There was never any explanation of what was done for each.
I can't see what is occurring in Figures 2/3/4/5 because the data points overlap each other so much.
The scale on the qRT-PCR graphs is not shown in a conventional way.
No explanation is given for why supernatant and cell lysate viral RNA levels are both provided, nor are the results presented in the appropriate context.
Reviewer 2 Report
The article “Computational Designing and Experimental Evaluation of 2 MERS-CoV-siRNAs in Selected Cells” describes the computational selection of twenty-one miRNA against MERS-CoV. The authors experimentally evaluate these siRNAs against the target pathogen in a set of cell lines. They identify virus inhibition efficacy of these potential siRNAs by the Ct value of quantitative real-time PCR.
The primary issue with the manuscript is Figure 4 and Figure 5 where Ct values of negative controls are higher than 58. A typical RT-PCR assay will have a maximum of 40 to 45 thermal cycles. The citation for the protocol is Azhar et al., 2014 which further cite the following paper Corman, V. M., et al. "Assays for laboratory confirmation of novel human coronavirus (hCoV-EMC) infections." Eurosurveillance 17.49 (2012): 20334. The protocol shows 45 thermal cycles.
Given the discrepancy, the manuscript may be rejected.
Azhar, E. I.; El-Kafrawy, S. A.; Farraj, S. A.; Hassan, A. M.; Al-Saeed, M. S.; Hashem, A. M.; Madani, T. A., Evidence for camel-291 to-human transmission of MERS coronavirus. The New England journal of medicine 2014, 370 (26), 2499-505.
Reviewer 3 Report
Authors presented the work "Computational Designing and Experimental Evaluation of 2 MERS-CoV-siRNAs in Selected Cells"
In general the work sounds and it is well-written. Points of concern:
1. Intoduction: it could be valuable if you extend the concept of siRNA and the relevance for people that are not familiar with terms. Please include the extended name for siRNA, etc... you can also show other strategies such as CRISPR (briefly).
Please refer to other studies using those strategies in the same or other pathogens!
2a. Line 73: why "orf1ab" was the only selected molecule for searching siRNA? Please justify. Why other genes were not considered?
2b. Lines 78-95: Explanations are not part of Methods. MOve to Introduction or Discussion.
3. Line 113: cite the study (if any) or please indicate "data not published/shown"
4. Line 141: please indicate the assumptions you followed for the ANOVA test.
5. Which threshold was used for qPCR? Negative control had a CT=89, does that make sense?
6. RNA extraction in cell supernatant and cell lysate was the same for both procedures?
7. Table 3. I recommend you use a color code (or a heatmap) to show the impact (CT values) in a easier way!
8. Please column names in table 3 must be in same format with the same number of decimals.
9. Please indicate explicitly the limitations of the study.
10,. Please extend the fact that in silico approaches are common to screen possible strategies to overcome infections, showing examples in other models. For example (please include it): https://pubmed.ncbi.nlm.nih.gov/35155843/ and other similar studies to make emphasis regarding bioinformatic approaches and their usability!